# What happened during COVID-19 in African ICUs? An observational study of pulmonary co-infections, superinfections, and mortality in Morocco

Younes Aissaoui[1,2]*, Youssef Ennassimi[1], Ismail Myatt[1], Mohammed El Bouhiaoui[1], Mehdi Nabil[1], Mohammed Bahi[1], Lamiae Arsalane[3,4], Mouhcine Miloudi[3,4], Ayoub Belhadj[1,2]

1 COVID-19 Intensive Care Unit, Avicenna Military Hospital, Marrakech, Morocco, 2 Biosciences and Health Research Laboratory, Faculty of Medicine and Pharmacy, Cadi Ayyad University, Marrakech, Morocco, 3 Microbiology and Virology Department, Avicenna Military Hospital, Marrakech, Morocco, 4 Department of Microbiology, Faculty of Medicine and Pharmacy, Cadi Ayyad University, Marrakech, Morocco

* younes.aissaoui@uca.ma

**Data Availability Statement:** All relevant data are within the manuscript and its Supporting Information files.

## Abstract

### Background

There is a growing literature showing that critically ill COVID-19 patients have an increased risk of pulmonary co-infections and superinfections. However, studies in developing countries, especially African countries, are lacking. The objective was to describe the prevalence of bacterial co-infections and superinfections in critically ill adults with severe COVID-19 pneumonia in Morocco, the micro-organisms involved, and the impact of these infections on survival.

### Methods

This retrospective study included severe COVID-19 patients admitted to the intensive care unit (ICU) between April 2020 and April 2021. The diagnosis of pulmonary co-infections and superinfections was based on the identification of pathogens from lower respiratory tract samples. Co-infection was defined as the identification of a respiratory pathogen, diagnosed concurrently with SARS-Cov2 pneumonia. Superinfections include hospital-acquired pneumonia (HAP) and ventilator-associated pneumonia (VAP). A multivariate regression analysis was performed to identify factors independently associated with mortality.

### Results

Data from 155 patients were analyzed. The median age was 68 years [62–72] with 87% of patients being male. A large proportion of patients (68%) received antibiotics before ICU admission. Regarding ventilatory management, the majority of patients (88%) underwent non-invasive ventilation (NIV). Sixty-five patients (42%) were placed under invasive mechanical ventilation, mostly after failure of NIV. The prevalence of co-infections, HAP and VAP was respectively 4%, 12% and 40% (64 VAP/1000 ventilation days). The most isolated

**Funding:** The author(s) received no specific funding for this work.

**Competing interests:** The authors have declared that no competing interests exist.

pathogens were *Enterobacterales* for HAP and *Acinetobacter* sp. for VAP. The proportion of extra-drug resistant (XDR) bacteria was 78% for *Acinetobacter* sp. and 24% for *Enterobacterales*. Overall ICU mortality in this cohort was 64.5%. Patients with superinfection showed a higher risk of death (OR = 6.4, 95% CI: 1.8–22; p = 0.004).

## Conclusions

In this single-ICU Moroccan COVID-19 cohort, bacterial co-infections were relatively uncommon. Conversely, high rates of superinfections were observed, with an increased frequency of antimicrobial resistance. Patients with superinfections showed a higher risk of death.

## Introduction

COVID-19 critically ill patients have an increased risk of pulmonary co-infection [1–3] and superinfection [3–8]. Co-infections are considered community–acquired pneumonia (CAP) and are provoked by respiratory flora diagnosed during the first 24 to 48 hours of hospital admission [9]. Co-infections may also be caused by intracellular pathogens [10]. Superinfections are hospital–acquired pulmonary infections occurring more than 48 hours after hospital admission [9, 11]. The frequency of superinfections among severe COVID-19 patients is related to multiple factors: prolonged ICU length of stay and prolonged mechanical ventilation, the use of immunomodulatory drugs like steroids and anti-interleukins; and the frequent prescription of unjustified broad-spectrum antibiotics [12–14]. The excessive use of antibiotics is accompanied by the emergence and diffusion of multi-drug resistant (MDR) pathogens.

There is more and more data about the prevalence of co-infections and superinfections among severe COVID-19 patients. However, studies in developing countries, especially African countries, are lacking [15]. In Morocco, there have been more than one million and a quarter of confirmed cases with more than 16,000 deaths [16]. Despite the lack of scientific evidence, the Moroccan ministry of health has adopted chloroquine and azithromycin as antiviral drugs [17]. Moreover, a massive prescription of antibiotics in COVID-19 patients was observed worldwide during this pandemic. Similar to other countries on the African continent, there is a data gap regarding COVID-19 critically ill patients in Morocco, the prevalence and microbiology of co-infections and superinfections, and patients' outcomes [15].

The aim of this study was to determine the prevalence of bacterial pulmonary co-infections and superinfections in severe COVID-19 pneumonia in a Moroccan ICU, the micro-organisms involved, and the impact of these infections on survival.

## Methods

### Study design

This observational retrospective study was performed in the COVID-19 intensive care unit (ICU) of Avicenna Military Hospital, a university-affiliated hospital, located in the city of Marrakesh, Morocco. The COVID-19 ICU was an open eight-bed ICU. All critical care beds were equipped with invasive and non-invasive mechanical ventilation. Renal replacement therapy consisting of intermittent hemodialysis was also available. Two senior intensivists were in charge of patients, supported by junior doctors during night shifts. The nurse-to-patient ratio varied between 1:2 and 1:4. The study was approved by the ethical committee of Cadi Ayyad University, which waived the need for patient-informed consent (N°04/2022). This study adheres to the STROBE statement.

## Patient selection

We included patients aged 18 years or over who were admitted to the ICU for severe or critical COVID-19 pneumonia between April 2020 and April 2021. To define severe or critical COVID-19 pneumonia, pulse oximetry (SpO2) < 90% on room air, severe respiratory distress, respiratory rate > 30 breaths/min [18, 19], or acute respiratory distress syndrome (ARDS) [20] were used. If arterial blood gas was not available, the SpO2/FiO2 index was used. The WHO guidelines consider the threshold of SpO2/FiO2<315 to be equivalent to the PaO2/FiO2 <300 mmHg to define ARDS [19]. Patients discharged in less than 48 hours and patients admitted for reasons other than severe or critical pneumonia were excluded. Fungal and viral co-infections or superinfections were also excluded.

## Microbiological testing

SARS-CoV2 infection was confirmed using reverse transcriptase-polymerase chain reaction (RT-PCR) performed on a nasopharyngeal swab or lower respiratory tract secretions if the patient was on invasive mechanical ventilation.

Microbiological testing for co-infections and superinfections was conducted if there was a clinical, biological, and/or radiological suspicion. Patients with purulent sputum, elevated procalcitonin or neutrophil levels, or lobal or segmental opacification on a chest CT scan were suspected of having pulmonary co-infections. Superinfections were suspected in patients with clinical deterioration (worsening of hypoxemia, reappearance of fever, purulent and increased respiratory secretions, sepsis or septic shock), apparition of a new infiltrate on pulmonary imaging, or increased inflammatory markers.

The microbiological diagnosis was based on the isolation of pathogens from blood or lower respiratory tract samples culture. The latter included sputum for spontaneously breathing patients and mini-bronchoalveolar lavage (BAL) for intubated patients. In our ICU, mini-BAL is routinely obtained within 24–48 hours after tracheal intubation or when ventilator-associated pneumonia (VAP) is suspected. The mini-BAL was performed with instillation and aspiration of 20 ml of saline without bronchoscopy [21]. The diagnostic thresholds for mini-BAL and sputum culture were $10^4$ CFU/mL and $10^5$ CFU/mL, respectively [22–24]. Blood culture was considered for diagnosis only if the respiratory sample identified the same pathogen or if there were no other compatible infectious sites (urine, catheter, etc.). Culture results were reviewed by an intensivist and a microbiologist to exclude results with contamination or colonization. In respiratory samples, coagulase-negative staphylococci, and non-pneumococcal streptococci were not considered relevant pathogens. Also, skin contaminants were not considered in blood culture results.

Pathogen susceptibility was interpreted according to the current European Committee on Antimicrobial Susceptibility Testing (EUCAST) guidelines. Multidrug-resistance (MDR), extensively drug-resistance (XDR), and pandrug-resistance (PDR) were defined according to the current consensus [25]. Additionally, a multiplex respiratory PCR (MR-PCR) was performed: BioFire® FilmArray® Pneumonia Plus V2.0 Panel (Biomérieux, Marcy-l'étoile, France). This MR-PCR detects nucleic acids from 9 viruses, 15 bacteria with a semiquantitative value, 3 atypical bacteria, and 7 antimicrobial resistance genes. In our unit, the practice of MR-PCR is restricted to mini-BAL. The same sample that was used for classic microbiological testing was concurrently used for MR-PCR.

## Definition of endpoints

The primary endpoint was the prevalence of laboratory-confirmed pulmonary co-infection and superinfection. Co-infections were classified as CAPs [22]. It was defined as the identification of bacterial respiratory pathogens, diagnosed concurrently with SARS-Cov2 pneumonia, using

MR-PCR or microbiological cultures from lower respiratory secretions (sputum or mini-LBA), obtained during the first 48 h of hospital admission. Pulmonary superinfection was defined as a hospital-acquired infection occurring more than 48 hours after hospital admission for SARS-Cov2 pneumonia. Superinfections include hospital-acquired pneumonia (HAP) and ventilator-associated pneumonia (VAP) defined according to current guidelines [23]. HAP and VAP were defined as pneumonia occurring 48 hours or more after admission or endotracheal intubation, respectively.

The secondary endpoints were the bacteria isolated, their susceptibilities to antibiotics, and the impact of these infections on the clinical outcome of patients. The resistance profile was defined as multidrug resistant (MDR) if the microorganisms were resistant to 1 drug in at least 3 classes of antibiotics, extensively drug-resistant (XDR) if resistant to $\geq 1$ drug in all but $\leq 2$ classes of antibiotics, and pandrug-resistant (PDR) if non-susceptible to all agents in all anti-microbial categories [25].

### Data collection

Patients' demographics, comorbidities, severity scores, exposure to antibiotics in the previous 30 days, hydroxychloroquine use before ICU admission, laboratory tests, percentage of affected lungs on first CT scan [26], therapeutic management, the need for non-invasive or invasive ventilation, and outcomes (duration of mechanical ventilation, ICU length of stay, in-hospital mortality) were collected from patients' files and hospital electronic records.

### Statistical analysis

Categorical variables are reported as absolute numbers (percentages) and continuous variables as median (interquartile). The Mann-Whitney U-test or the Chi-square test were used to compare differences between groups as appropriate. To evaluate the impact of superinfection on patients' outcomes, different factors, including superinfection, were compared between survivors and non-survivors. A multivariate regression analysis was performed to identify factors independently associated with mortality. The variables with a univariate $P < 0.05$ were entered into a logistic regression prediction model constructed using a forward stepwise procedure. Statistical significance was established at $P < 0.05$. The reported P values are two-sided. Statistical analyses were performed using SPSS version 25.0 (IBM®, Armonk, USA).

## Results

During the study period, 996 COVID-19 patients were admitted to our hospital. Among them, 183 patients (18.4%) were admitted to the ICU. Fig 1 shows the flow chart of the study. Ten patients admitted to the ICU for a diagnosis other than severe or critical COVID-19 pneumonia and 18 patients with a length of stay of under 48 hours were excluded. Therefore, 155 patients were analyzed. Non-bacterial pulmonary infections included 2 viral co-infections and 3 fungal VAPs which were excluded.

### Patient characteristics (Table 1)

The majority of patients were male (87%), with a median age of 68 years [IQR, 62–72]. The most common comorbidities were diabetes and arterial hypertension. High severity scores (APACHE2 and SOFA) were observed. The median duration of symptoms before ICU hospitalization was 7 days [IQR, 6.75–10]. More than half (55%) of the patients were admitted from the emergency department or lower-intensity wards. The median duration of hospitalization before ICU admission was 4 days [IQR, 2–6]. A large proportion of patients received antibiotics before ICU admission (58%), with third generation cephalosporins (3GC) and

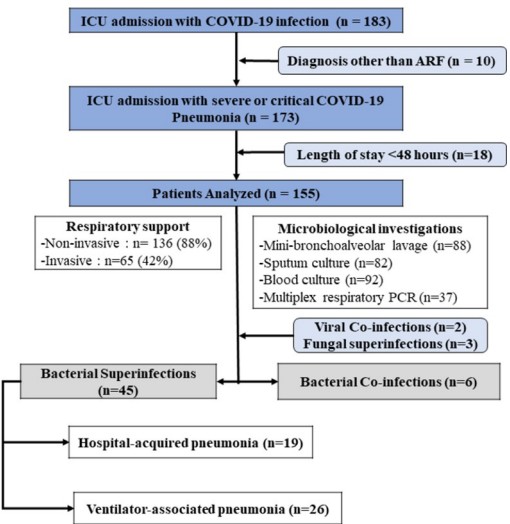

**Fig 1. Flow chart diagram.** ICU: intensive care unit, COVID-19: coronavirus disease 2019. * admission causes other than severe or critical COVID-19 pneumonia: acute kidney injury (n = 3) [hypovolemic Shock, acute pancreatitis, acute limb ischemia], non-traumatic coma (n = 2) [acute ischemic stroke, status epilepticus], cardiogenic choc (n = 2) [acute myocarditis, acute coronary syndrome], postoperative management of cardiac surgery (n = 1), severe metabolic disturbance (n = 2) [diabetic ketoacidosis]. † Viral co-infections were due to the influenza A virus in one patient and the human rhinovirus in another. Fungal superinfections were ventilator-associated pneumonia due to *Aspergillus fumigatus* in two patients and *Cryptococcus neoformans* in one patient.

azithromycin being the most prescribed antibiotics. A quarter of the patients included had hydroxychloroquine prescriptions before ICU admission.

This cohort had a severe respiratory condition, with a median PaO2/FiO2 ratio of 94 mmHg [80–118] and a median percentage of affected lungs on the first CT scan of 75% [58–75]. The patients included were also characterized by a high inflammatory profile, as illustrated by the elevated value of CRP. The median CRP was 180 mg/l [91–264].

### Therapeutic management in ICU (Table 2)

After ICU admission, empiric antibiotics were prescribed or continued for suspicion of pulmonary co-infection in almost one-third of patients (28%); mostly 3rd generation cephalosporins (3GC) were eventually associated with antipneumococcic quinolones (levofloxacin or moxifloxacin). A pulmonary superinfection (VAP or HAP) was suspected in 44% of patients. A carbapenem or antipseudomonal 3GC in combination with an aminoglycoside or colistin was the most commonly prescribed regimen. Regarding ventilatory management, the majority of patients (88%) underwent non-invasive ventilation (NIV). In nearly half of them, NIV was considered as a ceiling of care. Sixty-five patients (42%) were placed under invasive mechanical ventilation (MV), mostly after the failure of NIV. None of the patients admitted received high-flow oxygen. Steroids were administered to 93% of patients, with methylprednisolone being the most prescribed steroid. Low molecular weight heparin (enoxaparin) was administered in 98% of patients. Thirty-five patients (23%) received therapeutic doses of enoxaparin. Among them, 16 patients had therapeutic anticoagulation for a venous or arterial thrombotic event.

### Prevalence of pulmonary co-infections, superinfections and pathogens identified (Table 3)

Among the 155 included patients, 92 blood cultures, 82 sputum cultures, and 88 mini-BALs were achieved. Among the mini-BAL samples, 37 were also examined by multiplex

**Table 1. Characteristics of critically ill COVID-19 patients.**

| Patients' characteristics (n = 150) | |
|---|---|
| Age (years) | 68 (62–72) |
| Male / female [n (%)] | 135 (87%) / 20 (13%) |
| Charlson comorbidity index | 3 [3–4.25] |
| Comorbidities | |
| Diabetes [n (%)] | 56 (36%) |
| Arterial hypertension [n (%)] | 48 (31%) |
| Cardiopathy [n (%)] | 45 (29%) |
| Chronic lung disease [n (%)] | 34 (22%) |
| Cerebrovascular diseases [n (%)] | 3 (2%) |
| Obesity (body mass index>30 Kg/m$^2$) [n (%)] | 41 (26%) |
| Chronic kidney disease [n (%)] | 8 (5%) |
| SOFA on admission | 5 [4–5.75] |
| APACHE II on admission | 19 [16–20] |
| Data before ICU admission | |
| Symptoms duration before ICU hospitalization (days) | 7 [6.75–10] |
| Hospitalization in the wards before ICU admission [n (%)] | 85 (55%) |
| Hospitalization duration wards before ICU admission (days) | 4 [2–6] |
| Antibiotics before ICU admission [n (%)] | 90 (58%) |
| Amoxicillin-Clavulanate [n (%)] | 12 (18%) |
| Third generation cephalosporin (3GC) [n (%)] | 45 (66%) |
| Azithromycin [n (%)] | 90 (58%) |
| Fluroquinolones (FQ) [n (%)] | 7 (10%) |
| Antibiotic combination (3GC+FQ) [n (%)] | 4 (6%) |
| Antibiotherapy duration before ICU admission (days) | 4 [2–5.5] |
| Hydroxychloroquine [n (%)] | 37 (24%) |
| Indicators of respiratory severity on ICU admission | |
| Respiratory rate (breath/min) | 35 [32–40] |
| SpO$_2$/FiO$_2$ ratio | 80 [75–95] |
| PaO$_2$/FiO$_2$ ratio (mmHg) | 94 [80–118] |
| Percentage of affected lungs on the first CT scan (%) | 75 [58–75] |
| Baseline inflammatory and coagulation biomarkers | |
| Leukocytes cell count (10$^3$/ mm$^3$) | 11.9 [8.2–15.6] |
| Neutrophils cell count (10$^3$/ mm$^3$) | 10.5 [6.8–13.8] |
| Lymphocytes cell count (10$^3$/ mm$^3$) | 0.8 [0.6–1.2] |
| C-reactive protein (mg/L) | 179 [91–264] |
| Procalcitonin (ng/mL) | 0.32 [0.14–0.98] |
| Ferritin (ng/ml) | 750 [440–1200] |
| Lactate dehydrogenase (LDH) | 520 [360–727] |
| Fibrinogen (g/L) | 4.5 [3–5.9] |
| Platelet count (10$^3$/ mm$^3$) | 223 [160–336] |
| D-Dimer (μg/mL) | 2.2 [1.5–4.2] |

Continuous variables are expressed as median [Interquartile range] and categorical variables as numbers (percentages), ICU: intensive care unit, SOFA: Sequential Organ Failure Assessment, APACHEII: Acute Physiology and Chronic Health Evaluation II, COPD: chronic obstructive pulmonary disease, COVID-19: coronavirus disease.

PCR. Co-infection was observed in 6 patients, resulting in a prevalence of 4% (Table 3). The diagnosis was based on sputum culture in 3 patients, on mini-BAL culture in 2

**Table 2. Therapeutic management and outcomes of critically ill COVID-19 patients (n = 155).**

| | |
|---|---|
| Empiric antibiotics for co-infections' suspicion [n (%)] | 40 (28%) |
| Third generation cephalosporin* | 14 |
| Third generation cephalosporin + antipneumococcic fluoroquinolone | 15 |
| Amoxicillin-clavulanate | 8 |
| Ertapenem | 3 |
| Empiric antibiotics for superinfections' suspicion [n (%)] | 68 (44%) |
| Third generation cephalosporin | 18 |
| Third generation cephalosporin + fluroquinolone | 9 |
| Antipseudomonal third generation cephalosporin (AP3GC) † | 6 |
| Carbapenem (imipenem, meropenem) | 14 |
| Aminoglycoside or colistin + carbapenem or AP3GC | 21 |
| Ventilation management | |
| Non-invasive ventilation [n (%)] | 136 (88%) |
| Non-invasive ventilation as ceiling of care [n (%)] | 71 (46%) |
| Invasive ventilation [n (%)] | 65 (42%) |
| Proning in patient with invasive ventilation [n (%)] | 38 (58%) |
| Vasopressors and or inotropes | 62 (40%) |
| Renal replacement therapy (hemodialysis) | 19 (12%) |
| Immunomodulatory treatment | |
| Steroids [n (%)] | 144 (93%) |
| Methylprednisolone (40 mg / day) [n (%)] | 99 (69%) |
| Dexamethasone (6 mg / day) [n (%)] | 29 (20%) |
| Hydrocortisone (150–200 mg / day) [n (%)] | 16 (11%) |
| Anti-interleukin 6 (Tocilizumab) [n (%)] | 3 (4%) |
| Anticoagulants [n (%)] | 152 (98%) |
| Standard prophylactic dose (enoxaparin ≤4000 IU/d) [n (%)] | 30 (20%) |
| Intermediate dose (enoxaparin 100 IU/Kg/d) [n (%)] | 87 (57%) |
| Therapeutic dose (enoxaparin 100 IU/Kg /12h) [n (%)] | 35 (23%) |
| Patient's outcome | |
| Duration of NIV (days) | 5 [3–7] |
| Weaning from NIV [n (%)] | 38 (53%) |
| Day of invasive ventilation—ICU admission (days) | 4 [3–7] |
| Duration of invasive ventilation (days) | 6 [3–9.5] |
| Weaning from invasive ventilation [n (%)] | 6 (9%) |
| Length of ICU stay (days) | 9 [6–12] |
| Survival in patients with NIV as ceiling of care [n (%)] | 38 (53%) |
| Survival in patient with invasive mechanical ventilation [n (%)] | 5 (8%) |
| Survival in the overall cohort [n (%)] | 55 (35.5%) |
| Causes of deaths [n (%)] | |
| Refractory respiratory failure | 45 (45%) |
| Septic shock | 38 (38%) |
| Multiple organ failure | 9 (6%) |
| Others causes | 11 (11%) |

Continuous variables are expressed as median [Interquartile range] and categorical variables as numbers (percentages), ICU: intensive care unit,

*only ceftriaxone was available,

†ceftazidime.

**Table 3. Micro-organisms responsible for co-infection and superinfection in critically ill COVID-19 patients (n = 155).**

| Pulmonary co-infection, n = 6 | |
|---|---|
| Gram-positive (n = 3) | *Staphylococcus aureus MS* (n = 2) |
| | *Staphylococcus aureus MR* (n = 1) |
| Gram-negative (n = 3) | *Haemophilus influenzae* (n = 1) |
| | *Proteus* sp. (n = 1) |
| | *Klebsiella pneumoniae* (n = 1) |
| **Pulmonary superinfection, n = 45** | |
| **Hospital-acquired pneumonia, n = 19** | |
| Gram-negative (n = 14) | *Klebsiella pneumoniae* (n = 7) |
| | *Acinetobacter Baumanii* (n = 3) |
| | *Enterobacter cloacae* (n = 2) |
| | *Pseudomonas aeruginosa* (n = 1) |
| | *Proteus mirabilis* (n = 1) |
| Gram-positive (n = 3) | *Staphylococcus aureus MR* (n = 2) |
| | *Staphylococcus aureus MS* (n = 1) |
| Polymicrobial (N = 2) | *Enterobacter cloacae + Serratia marcescens* (n = 1) |
| | *Staphylococcus aureus MR + Klebsiella aerogenes* (n = 1) |
| **Ventilator-associated pneumonia, n = 26** | |
| Gram-negative (n = 16) | *Acinetobacter Baumanii* (n = 8) |
| | *Klebsiella pneumoniae* (n = 3) |
| | *Pseudomonas aeruginosa* (n = 3) |
| | *Klebsiella oxytoca* (n = 1) |
| | *Stenotrophomonas maltophilia* (n = 1) |
| Gram-positive (n = 2) | *Staphylococcus aureus MR* (n = 2) |
| Polymicrobial (n = 8) | *Acinetobacter baumanii + Klebsiella pneumoniae* (n = 3) |
| | *Acinetobacter Baumanii + Pseudomonas aeruginosa* (n = 2) |
| | *Acinetobacter Baumanii + Stenotrophomonas maltophilia* (n = 1) |
| | *Acinetobacter baumanii + Enterobacter cloacae* (n = 1) |
| | *Pseudomonas aeruginosa + Staphylococcus aureus MR* (n = 1) |

MS: methicillin-susceptible, MR: methicillin-resistant.

patients, and on blood culture in one patient. The bacterial micro-organisms isolated were *Staphylococcus aureus (n = 3)* for gram-positive pathogens and *Hemophilus influenza*, *Proteus* sp., and *Klebsiella pneumoniae* for gram-negative pathogens (n = 1 for each pathogen).

Forty-five pulmonary superinfections were observed (prevalence 27%), including 19 HAP and 26 VAP (Table 3). The prevalence of HAP was 12%. The documentation of HAP was performed with sputum culture in 12 patients, mini-BAL culture in 6 patients, and blood culture in one patient. The length of ICU stay at the time of HAP diagnosis was 9 days (IQR 5–12). The prevalence of VAP in ventilated patients was 40%, resulting in an incidence density (ID) of 64 VAP episodes / 1000 ventilation days. Two patients developed more than one VAP episode. The median duration of mechanical ventilation at the time of VAP identification was 8 (IQR 6–10) days. Gram-negative bacilli (GNB) were the predominantly identified bacteria in HAP, in particular *Klebsiella pneumoniae* (n = 7). For VAP, the spectrum of pathogens was also dominated by GNB, with *Acinetobacter baumanii* (*AB*) [n = 15] being the most isolated pathogen. About a third of VAPs were polymicrobial (n = 8). A positive PCR alone was used

to make the diagnosis in only 2 cases: a co-infection due to *Hemophilus influenza* and a VAP due to *Klebsiella oxytoca*.

Regarding the resistance profile observed in superinfections, 78% (n = 14) of *AB* isolates (n = 18) had an XDR profile (susceptibility to colistin only), while the rest of the isolates (22%) had an MDR profile (n = 4). All the *AB* isolates were resistant to carbapenems. Moreover, the multiplex PCR showed that 80% of *Acinetobacter* isolates exhibited the NDM resistance gene. *Enterobacterales* (n = 29) had an XDR profile in 24% (n = 7) of isolates and an MDR profile in 48% (n = 14) of isolates. 81% of *Klebsiella pneumonia* isolates were found to be resistant to third-generation cephalosporins, producing extended-spectrum beta-lactamases (ESBL), and 25% resistant to carbapenems. The multiplex PCR showed that 90% of *Klebsiella pneumonia* isolates exhibited the CTX-M gene.

## Factors associated with mortality (Table 4)

ICU mortality was 64.5%. In univariate analysis, factors associated with mortality were age, higher SOFA and APACHE2 scores, lower oxygenation index, higher values of CRP, procalcitonin, creatinine, and LDH. The need for mechanical ventilation and the occurrence of superinfection were also significantly associated with mortality. Multivariate analysis showed that a CRP value at admission > 179 mg/L (OR = 4.8, 95% CI: 1.7–13.5; p = 0.003), an age > 68 years (OR = 4.3, 95% CI: 1.5–12; p = 0.007) and the occurrence of superinfection (OR = 6.4, 95% CI: 1.8–22; p = 0.004) were significantly associated with mortality.

## Discussion

To date, this is the first study reporting data about pulmonary co-infections and superinfections among severe COVID-19 in an African country. The prevalence of bacterial co-infections was low (4%). On the other hand, superinfections including HAP and VAP were common. The microorganisms identified in superinfections were predominantly *Enterobacterales* for HAP, in particular Klebsiella pneumoniae, and Acinetobacter sp. for VAP. There was a worryingly increased proportion of antimicrobial resistance. Moreover, this Moroccan ICU cohort showed a high mortality rate, and death was significantly associated with the occurrence of superinfections.

The low rate of co-infection observed in this study is concordant with the literature. Indeed, two meta-analyses found a similarly low rate of co-infection among COVID-19 patients. However, the majority of included studies confused community-acquired (co-infections) and hospital-acquired infections (superinfections) [9, 27]. A study of 254 critically ill COVID-19 patients from England, with a comparable rate of patients under MV, reported a co-infection prevalence of 5% [28]. In contrast, studies involving the most severely ill patients (i.e., those on mechanical ventilation) revealed a relatively high co-infection prevalence ranging from 21% to 28% [1–3]. This variability in co-infection rates may be explained by the severity of patients included, the nature of respiratory samples collected (sputa vs invasive trachea-bronchial aspirates or BAL), the proportion of patients receiving prior antibiotics, and the sensitivity of diagnostic techniques (conventional culture vs molecular methods). The use of multiplex PCR nearly doubled the detection of co-infections among critically ill COVID-19 ICU patients [29, 30]. The majority of the studies cited above reported a high rate of antimicrobial use in up to 90% of patients. The commonest pathogens identified were Staphylococcus aureus, Streptococcus pneumoniae, Haemophilus influenzae, and *Enterobacterales*, [1–3, 28–30]. Some reports showed that patients with COVID-19 may also have co-infections caused by intracellular agents [10].

The prevalence of pulmonary superinfections in this study was high (27%). In particular, the prevalence of VAP was 40%. The different cohorts published reported that a VAP

**Table 4. Comparison between survivors and non survivors in critically ill COVID-19 patients.**

|  | Survivors (n = 55) | Non survivors (n = 100) | P |
|---|---|---|---|
| Age (years) | 64 (59–70) | 68 [64–74] | <0.001 |
| Male | 51 (93%) | 84 (84%) | 0.140 |
| Charlson comorbidity index | 3 [2–4.25] | 4 [3–4.75] | 0.092 |
| Diabetes | 19 (34%) | 37 (37%) | 0.861 |
| Arterial hypertension | 10 (18%) | 38 (38%) | 0.030 |
| Cardiopathy | 14 (26%) | 31 (31%) | 0.579 |
| Chronic lung disease | 9 (16%) | 25 (25%) | 0.231 |
| Obesity (body mass index>30 Kg/m$^2$) | 17 (31%) | 24 (24%) | 0.439 |
| SOFA on admission | 7 [6–7] | 9 [8–9] | <0.001 |
| APACHE II on admission | 15 [15–17] | 19 [18–21] | <0.001 |
| Symptoms duration before ICU (days) | 10 [7–12] | 7 [6–10] | 0.036 |
| Hospitalization before ICU admission | 33 (60%) | 52 (52%) | 0.399 |
| Antibiotics > 24 h before ICU admission* | 36 (65%) | 54 (54%) | 0.178 |
| Respiratory rate (breath/min) | 35 [30–35] | 36 [32.5–40] | 0.01 |
| SpO$_2$/FiO$_2$ ratio | 85 [80–87] | 80 [70–81.5] | 0.001 |
| Percentage of affected lungs on the first CT scan (%) | 75 [50–75] | 75 [67.5–75] | 0.052 |
| Leukocytes cell count (10$^3$/ mm$^3$) | 10.8 [8.3–13.5] | 12.5 [8.1–16.2] | 0.394 |
| Neutrophils cell count (10$^3$/ mm$^3$) | 9.8 [7–11.5] | 11.1 [6.3–4.3] | 0.270 |
| Lymphocytes cell count (10$^3$/ mm$^3$) | 0.8 [0.6–1.2] | 0.8 [0.5–1] | 0.156 |
| C-reactive protein (mg/L) | 126 [55–182] | 210 [116–171] | 0.005 |
| Creatinine (μmol/L) | 107 [82–147] | 83 [74–103] | <0.001 |
| Procalcitonin (ng/mL) | 0.13 [0.08–0.24] | 0.5 [0.2–1.2] | 0.009 |
| Ferritin | 673 [347–981] | 756 [646–1230] | 0.173 |
| LDH | 405 [250–425] | 607 [480–803] | 0.02 |
| Fibrinogen | 4.8 [2.4–5.8] | 4.5 [3.3–5.9] | 0.465 |
| D-Dimers | 2150 [1855–3855] | 2500 [1500–4250] | 0.839 |
| Non-invasive ventilation | 49 (89%) | 87 (87%) | 0.810 |
| Invasive ventilation | 5 (9%) | 60 (60%) | <0.001 |
| Vasopressors and or inotropes | 4 (7%) | 58 (58%) | <0.001 |
| Renal replacement therapy (hemodialysis) | 2 (3.6%) | 17 (17%) | 0.015 |
| Steroids | 50 (91%) | 94 (95%) | 0.521 |
| ICU length of stay (days) | 8 [6–11] | 10 [6–14] | 0.490 |
| Co-infection † | 1 (2%) | 5 (5%) | 0.424 |
| Superinfection (HAP or VAP) | 5 (9%) | 38 (38%) | <0.001 |

Continuous variables are expressed as median [Interquartile range] and categorical variables as numbers (percentages), *: azithromycin included, ICU: intensive care unit, *SOFA*: Sequential Organ Failure Assessment, *APACHEII* Simplified Acute Physiology Score II, HAP: hospital acquired pneumonia, VAP: ventilatory acquired pneumonia.

developed in about half of invasively ventilated COVID-19 patients [3–8]. The highest prevalence (79%) was observed in the French study of Maes [8]. The hazard ratio for developing VAP in invasively ventilated COVID-19 patients ranged from 1.7 to 2.2 [4, 5, 8]. When reported to MV duration, the VAP rate found in our study reached unprecedented values with an ID of 64 VAP/1000 ventilator days. The ID varied in the studies cited above between 28 and 45/1000 ventilator days [3, 7, 31]. An ID rate of 45/1000 ventilator days was observed in a North American ICU [3]. These disparities in VAP incidence may be related to underdiagnosing in some studies (less sampling due to the fear of healthcare worker contamination), the

nature of respiratory samples (distal vs proximal), the diagnosis methods (PCR vs culture), and empiric antibiotics prior to sampling.

Several factors have been proposed to explain such a high rate of VAP, including disease and therapy-induced immune impairment; prolonged MV; understaffing or inexperienced staff; widespread prescription of broad-spectrum antibiotics; and low use of preventive bundles [12–14]. During the pandemic, the COVID-19 ICU described in this study was particularly understaffed, with a nurse-to-patient ratio reaching up to 1:4. Moreover, a large proportion of nurses involved in the COVID-19 ICU had limited or no critical care training. Regarding the use of immunosuppressive therapy, 93% of patients in this cohort received corticosteroids. In a multicenter prospective study including more than two thousand severe COVID-19 patients, Reyes found that dexamethasone was associated with an increased risk of pulmonary superinfection with an odds ratio of 1.64 [32]. Anti-interleukins were rarely used in this cohort due to their high cost. Lastly, some logistical factors could also explain this high frequency of superinfections in our study, since many medical supply shortages happened during this period (hand disinfectants, suction catheters, gloves, etc).

The distribution of pathogens identified in HAP was comparable to the literature [11, 31] and consisted largely of GNB, particularly *Klebsiella pneumoniae* (Table 3). These pathogens are usually associated with HAP and VAP and are not specific to COVID-19 cohorts [4, 5, 8, 12, 13]. Interestingly, *Klebsiella pneumoniae* showed a high resistance profile, with 81% of 3GC resistance and 25% of carbapenem-resistance. The rate of 3GC-resistance was higher than that reported in Covid-19 ICU European cohorts (half of *Enterobacterales* resistant to 3GC) with a high rate of ESBL [7, 8, 12]. The high rate of ESBL is consistent with our finding since the CTX-M gene was detected in 90% of *Klebsiella* isolates in our study.

A remarkable result regarding VAP microbiology was the predominance of *Acinetobacter baumanii* (AB). The latter was identified as the sole bacteria isolated or associated with another microorganism (polymicrobial VAP) in 58% of the 28 VAP episodes. The predominance of this non-fermenting GNB as a VAP pathogen was not reported in any of the studies mentioned above, including studies from Europe, the United States, and China. In the multicenter European study of Rouzé, *AB* represented only 7% of the microorganisms responsible for ventilator-associated lower respiratory tract infections [4]. On the contrary, a retrospective Iranian study reported that XDR *AB* was responsible for 90% of VAP in COVID-19 patients [33]. Similar to our study, all *AB* strains were carbapenem-resistant. However, carbapenem resistance was not driven by a metallo-beta-lactamase enzyme. On the opposite, the *AB* strains identified in our study were metallo-beta-lactamase-producing strains since the NDM gene was identified in 80% of *AB* strains. Another noticeable difference is the 100% colistin susceptibility in our study versus 48% in the Iranian study. This ICU outbreak of *AB* in our study could be explained by several factors, such as the open plan of the COVID-19 ICU, the improper environmental cleaning due to a lack of cleaning staff and overcrowding, and increased patient-to-patient transmission from inexperienced ICU personnel [34]. During the COVID-19 pandemic, there was a spike in AB healthcare- associated infections, primarily lower respiratory tract infections, in a number of ICU and non-ICU settings [35]. This AB outbreak inside the COVID-19 outbreak underlines the importance of appropriate prevention and control measures.

This cohort of COVID-19 critically ill patients showed a high mortality rate of 64.5%. The latter is higher than that reported from studies done in Asia, Europe, and North and South America [3–6, 8]. A meta-analysis examining the mortality of severe COVID-19 patients reported that the overall global mortality was 31.5% (95% CI 27.5%–35.5%) [15]. This metanalysis, which accompanied a multicenter African study, investigated the outcomes of more than 3,000 critically ill African COVID-19 patients and found a mortality rate of 48% (95% CI 46–50); lower than that reported in our study [15]. Nevertheless, the authors themselves

recognized that this African cohort was younger (median age of 56 vs 68 years in our cohort) and likely had fewer comorbidities [15]. COVID-19 patients' mortality is higher in older and comorbid patients [6]. The severity of patients included is probably another explanation for this difference in outcomes (higher SOFA scores in our study).

The independent factors associated with death in our study were age, the CRP value at admission, which is a surrogate of the severity of the disease, and the occurrence of pulmonary superinfection. The latter is not always linked to poorer outcomes in severe COVID-19 patients [3, 4, 6, 12, 28]. The multicenter study by Baskaran showed that ICU patients developing superinfections (VAP and HAP) were more likely to die. The crude OR was 1.78 (95% confidence interval: 1.03–3.08, P = 0.04) [28]. On the contrary, the development of VAP was not significantly associated with death in other studies [3, 6]. It is not surprising that superinfection was associated with mortality in our study since responsible microorganisms were characterized by a high resistance profile. Indeed, carbapenem-resistant *AB* and 3GC-resistant *Klebsiella pneumoniae* are among the six leading pathogens for death associated with antimicrobial resistance [36].

Our study has some limitations. First, this is a single-center retrospective study, which may not reflect the prevalence and ecology of co-infections and superinfections in other Moroccan ICUs. However, this study will probably contribute to filling the data gap in low-income settings, particularly in Africa. Second, this is a study reporting only microbiologically documented infections. Microbiological tests were not performed systematically but only in patients with clinical suspicion of infection. Moreover, a significant proportion of patients received antibiotics prior to microbiological sampling (more than half of the patients received antibiotics before ICU admission). These factors may have led to an underestimation of both co-infection and superinfection rates. Finally, mortality in this patient's series cannot be simply related to superinfection and MDR pathogens. The cause of death is clearly multifactorial, involving patient age, comorbidities, COVID-19 severity, and therapeutic strategy, particularly ventilatory management. The majority of patients were exposed to NIV (88%), with a high failure rate (47%). Indeed, a recent review including 4776 patients found that NIV failed in half of the patients and was associated with high mortality [37].

In conclusion, this study showed a low rate of bacterial pulmonary co-infection in critically ill COVID-19 patients. On the contrary, superinfections, in particular VAP, were common. The main micro-organism identified in VAPs was XDR *Acinetobacter* baumanii. *Enterobacterales*, especially 3GC-resistant *Klebsiella pneumoniae*, were predominantly related to HAP. The mortality of this ICU cohort was high and significantly associated with superinfection occurrence. These results underline the urgent need for antimicrobial stewardship policies and for trained ICU staff in developing countries such as Morocco.

## Supporting information

**S1 Data.**
(XLSX)

## Acknowledgments

We would like to thank Dr Youssef Ghazi for English proofreading. We are grateful to all the personnel who were involved in the care of patients.

## Author Contributions

**Conceptualization:** Younes Aissaoui, Ayoub Belhadj.

**Data curation:** Younes Aissaoui, Youssef Ennassimi, Ismail Myatt, Mohammed El Bouhiaoui, Mehdi Nabil, Lamiae Arsalane, Mouhcine Miloudi.

**Formal analysis:** Younes Aissaoui.

**Funding acquisition:** Younes Aissaoui.

**Investigation:** Younes Aissaoui, Youssef Ennassimi, Ismail Myatt, Mohammed El Bouhiaoui, Mehdi Nabil.

**Methodology:** Younes Aissaoui, Youssef Ennassimi, Ayoub Belhadj.

**Project administration:** Younes Aissaoui, Youssef Ennassimi.

**Software:** Younes Aissaoui, Youssef Ennassimi, Lamiae Arsalane, Mouhcine Miloudi.

**Supervision:** Younes Aissaoui, Ayoub Belhadj.

**Validation:** Younes Aissaoui, Mohammed Bahi.

**Writing – original draft:** Younes Aissaoui, Youssef Ennassimi, Ismail Myatt, Mohammed Bahi, Ayoub Belhadj.

**Writing – review & editing:** Younes Aissaoui, Lamiae Arsalane, Mouhcine Miloudi, Ayoub Belhadj.

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
