## [Decision Letter · Decision Letter 0]

22 Sep 2022

PONE-D-22-23389What happened during COVID-19 in African ICUs? An observational study of pulmonary co-infections, superinfections, and mortality in Morocco.PLOS ONE

Dear Dr. Aissaoui,

Thank you for submitting your manuscript to PLOS ONE. After careful consideration, we feel that it has merit but does not fully meet PLOS ONE’s publication criteria as it currently stands. Therefore, we invite you to submit a revised version of the manuscript that addresses the points raised during the review process.

We look forward to receiving your revised manuscript.

Kind regards,

SHUI YEE LEUNG

Academic Editor

PLOS ONE

Journal Requirements:

Reviewers' comments:

Reviewer's Responses to Questions

**Comments to the Author**

1. Is the manuscript technically sound, and do the data support the conclusions?

Reviewer #1: Yes

Reviewer #2: Yes

2. Has the statistical analysis been performed appropriately and rigorously? 

Reviewer #1: Yes

Reviewer #2: Yes

3. Have the authors made all data underlying the findings in their manuscript fully available?

Reviewer #1: Yes

Reviewer #2: Yes

4. Is the manuscript presented in an intelligible fashion and written in standard English?

Reviewer #1: No

Reviewer #2: Yes

5. Review Comments to the Author

Reviewer #1: Authors performed an interesting study evaluating the prevalence of respiratory co-infections and superinfections in a cohort of COVID-19 ICU patients in Morocco and their impact on mortality.

Although several studies have described superinfections in critically ill patients, this study is of importance since presents data in an African country. Furthermore, authors presented data on co-infection rate.

Similar studies are welcome since they underline the need of following antimicrobial stewardship and infection control principles in order to reduce the rate of superinfections, which are mostly caused by MDR pathogens and have an important role in determining a worse outcome.

I have the following comments:

- Abstract

• It seems that the majority of patients had NIV (88%) but 42% also mechanical ventilation. I therefore assume that amongst patients initially treated with NIV, a quote was further treated with MV also. Please check and/or specify. The same in the text

• Authors should specify that results refer to a single ICU in Morocco

- extra-drug resistant (XDR): please check the abbreviation

- Please rename Enterobacteriaceae with Enterobacterales

- English language should be revised along the manuscript

- “Co-infections are considered community acquired pneumonia (CAP) and are provoked by respiratory flora diagnosed during the first 24 to 48 hours of hospital admission [9].”: Authors should also state/discuss that co-infections may be also caused by intracellular pathogens such as legionella, Chlamydia and/or Mycoplasma. Indeed, it has been demonstrated the role of Mycoplasma and Chlamydia as aetiological agents of co-infections during COVID19 (see Oliva et al, Co-infection of SARS-CoV-2 with Chlamydia or Mycoplasma pneumoniae: a case series and review of the literature. Infection. 2020 Dec;48(6):871-877. doi: 10.1007/s15010-020-01483-8. Epub 2020 Jul 28. PMID: 32725598; PMCID: PMC7386385.). Authors should also discuss these pathogens in the discussion part.

- “The Moroccan ministry of health has adopted chloroquine and azithromycin as antiviral drugs despite the lack of scientific evidence”: please add a ref

- “Moreover, a huge prescription of antibiotics in COVID-19 patients was also observed during this pandemic”: did the authors specifically refer to Morocco or in general? I would say in general, but please specify.

- Authors refer to co-infections and superinfections involving the lung, namely pneumonia: please specify it when referring to co-infections and, especially, superinfections

- Were patients admitted to the ICU directly from the ER or from different lower intensity wards? In the latter case, did authors consider superinfections developed only during the ICU stay or during the entire hospitalization?

- I would consider the provided definition refer mostly to superinfections rather than co-infection. Were tests for pathogens causing co-infections made in all the patients at ICU admission or only if there was a clinical suspicion of co-infections? Please specify.

- “The diagnostic thresholds for mini-BAL and sputum culture were 104 CFU/mL and 105 CFU/mL, respectively”: add a ref

- “In respiratory samples, Candida, coagulase-negative staphylococci, and nonpneumococcal streptococci were not considered relevant pathogens”: as stated in patient selection, authors excluded fungal pathogens. Therefore, I would also exclude Candida from this sentence

- How CT percentage involvement was measured? Please specify or insert a ref.

- Prior antibiotic exposure: did the authors intend during hospitalization or in the previous 30-d?

- Was in-hospital mortality the principal outcome? Please specify

- Please check all abbreviations

- “During the study period, 996 COVID-19 patients were admitted to our institution. Among them, 183 patients were admitted to the ICU”: please insert also the %

- Please change the word “incriminated”

- Overall along the result section: please insert the number of patients for the corresponding pathogens

- Please write bacteria correctly

- “The preponderance of this non-fermenting GNB was not reported in any of the studies above in Europe, North America, or China”: this sentence is not clear.

- Authors should also discuss the rise in the prevalence of Ab infections in the ICU during the COVID19 pandemic, which has been described in the literature

- Table1. Please add the unit of measure (ie years for age). Please add the first row with the total study population (n=155). I would not include azythromicin as an antiviral; rather, this is an antibiotic which has been used during COVID19 for its supposed action against SARS CoV2. The same in the text.

- Table3. Please check numbers (VAP due to GN seems to be 15)

- Table4. Please check the row co-infection

Reviewer #2: Dear Authors,

I commend your dedication to science and medicine in such a period of great strain for the critical care community. I read your paper with interest. I found it informative and valuable. Nevertheless, I have some comments for you. I think a minor revision is necessary to accept the paper on PLOS One.

Abstract:

Please add something regarding the statistical methods.

Please reformulate the phrase "Death was associated with superinfection." The sample size and methods do not allow you to demonstrate any association. Instead, you may just say, "patients with superinfection showed a higher risk of death."

Introduction:

Please introduce in the reference "Crit Care. 2022 Jun 13;26(1):176." regarding the increased risk of infection associated with corticosteroids.

Methods:

well done

Results:

well done

Discussion:

well done.

6. PLOS authors have the option to publish the peer review history of their article (what does this mean?). If published, this will include your full peer review and any attached files.

Reviewer #1: No

Reviewer #2: **Yes: **VITTORIO SCARAVILLI

---

## [Author Response · Author response to Decision Letter 0]

15 Oct 2022

Manuscript Number: PONE-D-22-23389

What happened during COVID-19 in African ICUs? An observational study of pulmonary co-infections, superinfections, and mortality in Morocco.

Dear Dr SHUI YEE LEUNG,

We sincerely thank you and the reviewers for your efforts to assess our manuscript and for giving us the opportunity to revise it. We think that the constructive comments and suggestions of the reviewers have markedly contributed to improving the quality and readability of our paper. We responded to all issues raised by the reviewers and made changes to the manuscript according to their comments. As suggested, these changes are outlined in the corrected manuscript. 

We hope that you and the reviewers will find our changes adequate and our paper acceptable for publication. We look forward to hearing from you.

Sincerely yours.

Younes Aissaoui on behalf of the authors.

Reviewers' comments:

Reviewer #1: Authors performed an interesting study evaluating the prevalence of respiratory co-infections and superinfections in a cohort of COVID-19 ICU patients in Morocco and their impact on mortality.

Although several studies have described superinfections in critically ill patients, this study is of importance since presents data in an African country. Furthermore, authors presented data on co-infection rate.

Similar studies are welcome since they underline the need of following antimicrobial stewardship and infection control principles in order to reduce the rate of superinfections, which are mostly caused by MDR pathogens and have an important role in determining a worse outcome.

R: We thank the reviewer for his/her valuable comments and his/her detailed and accurate revision. We have taken into account all the comments and have revised our manuscript accordingly. We hope that our manuscript will be significantly improved.

I have the following comments:

- Abstract

• It seems that the majority of patients had NIV (88%) but 42% also mechanical ventilation. I therefore assume that amongst patients initially treated with NIV, a quote was further treated with MV also. Please check and/or specify. The same in the text.

R: Almost all the patients who were placed under invasive mechanical ventilation underwent a trial of non-invasive ventilation before being intubated. This point has been clarified in the abstract: page 2, lines 37 – 39: “ the majority of patients (88%) underwent non-invasive ventilation (NIV). Sixty-five patients (42%) were placed under invasive mechanical ventilation, mostly after failure of NIV.”

 It was also clarified in the section result of the revised version of the manuscript: Page 11, lines 243 to 246: “Regarding ventilatory management, the majority of patients (88%) underwent non-invasive ventilation (NIV). In nearly half of them, NIV was considered as a ceiling of care. Sixty-five patients (42%) were placed under invasive mechanical ventilation (MV), mostly after failure of NIV.”

• Authors should specify that results refer to a single ICU in Morocco

R: As you suggested, we mentioned that it is a single center study: page 3, line 45: “In this single-center Moroccan ICU COVID-19 cohort,…”

- extra-drug resistant (XDR): please check the abbreviation

R: We checked the abbreviation of extra-drug resistant in the international expert proposal for standard definitions of acquired resistance (Magiorakos et al. Clin Microbiol Infect 2012; 18: 268–281). XDR is the correct definition.

- Please rename Enterobacteriaceae with Enterobacterales

R: As you recommended, we replaced "Enterobacteriaceae" with "Enterobacterales" throughout the manuscript.

- English language should be revised along the manuscript

R: As you requested, the manuscript was reviewed by a native English speaker.

- “Co-infections are considered community acquired pneumonia (CAP) and are provoked by respiratory flora diagnosed during the first 24 to 48 hours of hospital admission [9].”: Authors should also state/discuss that co-infections may be also caused by intracellular pathogens such as legionella, Chlamydia and/or Mycoplasma. Indeed, it has been demonstrated the role of Mycoplasma and Chlamydia as aetiological agents of co-infections during COVID19 (see Oliva et al, Co-infection of SARS-CoV-2 with Chlamydia or Mycoplasma pneumoniae: a case series and review of the literature. Infection. 2020 Dec;48(6):871-877. doi: 10.1007/s15010-020-01483-8. Epub 2020 Jul 28. PMID: 32725598; PMCID: PMC7386385.). Authors should also discuss these pathogens in the discussion part.

R: As you suggested, we discussed the role of intracellular pathogens as co-infecting agents in Covid-19 pneumonia and we cited the reference of Oliva et al.

-Section introduction: page 4, lines 67 - 68: “Co-infections may be also caused by intracellular pathogens.”

- Section discussion, page 16, line 348-349 : “Some reports showed that patients with COVID-19 may also have co-infections caused by intracellular agents.” 

- “The Moroccan ministry of health has adopted chloroquine and azithromycin as antiviral drugs despite the lack of scientific evidence”: please add a ref

R: As requested, the reference was added. Section introduction, page 4, Line 78, Reference 17.

Reference [17] : https: //www.covidmaroc.ma/Documents/2020/coronavirus/PS/CIR-protocole%20pec%20patients%20et%20leurs%20contacts%20et%20mises%20à%20jour%20des%20définitions.pdf.

- “Moreover, a huge prescription of antibiotics in COVID-19 patients was also observed during this pandemic”: did the authors specifically refer to Morocco or in general? I would say in general, but please specify.

R: The huge prescription of antibiotics refers to worldwide practice. We clarified it in the introduction, page 4, Line 79. “Moreover, worldwide, a huge prescription of antibiotics in COVID-19 patients was also observed during this pandemic.”

- Authors refer to co-infections and superinfections involving the lung, namely pneumonia: please specify it when referring to co-infections and, especially, superinfections

R: We specify it: , page 4, Line 82 “The aim of this study was to determine the prevalence of bacterial pulmonary co-infections and superinfections….”

- Were patients admitted to the ICU directly from the ER or from different lower intensity wards? In the latter case, did authors consider superinfections developed only during the ICU stay or during the entire hospitalization?

R: Thank you for this pertinent comment. More than half of patients (55%) were admitted from ERs or low-intensity wards. It is detailed in the section results, page 9, lines 194 -195. 

Regarding the second part of the question, the diagnosis of superinfection was made considering the entire hospitalization (not only the ICU stay).

- I would consider the provided definition refer mostly to superinfections rather than co-infection. Were tests for pathogens causing co-infections made in all the patients at ICU admission or only if there was a clinical suspicion of co-infections? Please specify.

R: During the initial phase of the pandemic, there was a confusion between pulmonary co-infections and superinfections. Most experts now agree that if the diagnosis is made within 2 days of COVID-19 hospital admission, these infections are defined as community-acquired co-infections. If diagnosis occurred 2 days after admission for COVID-19, these infections are defined as hospital-acquired superinfections. [Russell et al. Lancet Microbe 2021;2: e354–65] [Garcia-Vidal et la. Clinical Microbiology and Infection 2021].etc

 The tests were done only if there was a clinical suspicion of co-infection. This is specified in the section Methods, page 6, Lines 112 to 117. “Pulmonary co-infections were suspected in patient with purulent sputum production, elevated values of procalcitionin or neutrophile, lobal or segmental opacification on CT scan.”

- “The diagnostic thresholds for mini-BAL and sputum culture were 104 CFU/mL and 105 CFU/mL, respectively”: add a ref

R: We added the following references. 

[22] Metlay JP, Waterer GW, Long AC, Anzueto A, Brozek J, Crothers K, et al. Diagnosis and Treatment of Adults with Community-acquired Pneumonia. An Official Clinical Practice Guideline of the American Thoracic Society and Infectious Diseases Society of America. Am J Respir Crit Care Med. 2019 Oct 1;200(7):e45-e67. https://doi.org/10.1164/rccm.201908-1581ST. 

[23] Kalil AC, Metersky ML, Klompas M, Muscedere J, Sweeney DA, Palmer LB, et al. Management of Adults With Hospital-acquired and Ventilator-associated Pneumonia: 2016 Clinical Practice Guidelines by the Infectious Diseases Society of America and the American Thoracic Society. Clin Infect Dis. 2016 Sep 1;63(5):e61-e111. https://doi.org/10.1093/cid/ciw353. 

[24] Johansson N, Kalin M, Tiveljung-Lindell A, Giske CG, Hedlund J. Etiology of community-acquired pneumonia: increased microbiological yield with new diagnostic methods. Clin Infect Dis. 2010 Jan 15;50(2):202-9. https://doi.org/10.1086/648678. 

- “In respiratory samples, Candida, coagulase-negative staphylococci, and nonpneumococcal streptococci were not considered relevant pathogens”: as stated in patient selection, authors excluded fungal pathogens. Therefore, I would also exclude Candida from this sentence

R: We agree with the reviewer. It was deleted: page 6 , line 126.

- How CT percentage involvement was measured? Please specify or insert a ref.

R: CT percentage was measured according to the method described by Bernheim. The reference was added. Reference [26] : Bernheim A, Mei X, Huang M, Yang Y, Fayad ZA, Zhang N, et al. Chest CT Findings in Coronavirus Disease-19 (COVID-19): Relationship to Duration of Infection. Radiology. 2020 Jun;295(3):200463. https://doi.org/10.1148/radiol.2020200463. 

- Prior antibiotic exposure: did the authors intend during hospitalization or in the previous 30-d?

R: You are right. We meant prior antibiotic exposure in the previous 30-d. It was clarified: page 7, lines 152 : “comorbidities, severity scores, exposure to antibiotics in the previous 30 days, hydroxychloroquine use before ICU admission, …”

- Was in-hospital mortality the principal outcome? Please specify

R: you are also right. It was the in-hospital mortality. It was specified: page 7, line 155.

- Please check all abbreviations

R: we checked all he abbreviations as required.

- “During the study period, 996 COVID-19 patients were admitted to our institution. Among them, 183 patients were admitted to the ICU”: please iànsert also the %

R: It was inserted. Page 9, line 18 : “Among them, 183 patients (18.4%) were admitted to the ICU.”

- Please change the word “incriminated”

R: It was replaced by the word “identified”. Page 13 , line 266

- Overall, along the result section: please insert the number of patients for the corresponding pathogens

- Please write bacteria correctly

R: As you suggested, we inserted the patients ‘number (page 13) and corrected the word bacteria.

- “The preponderance of this non-fermenting GNB was not reported in any of the studies above in Europe, North America, or China”: this sentence is not clear.

R: We tried to make this sentence clearer. Page 17, lines 380-381 : “The predominance of this non-fermenting GNB as a VAP pathogen was not reported in any of the studies mentioned above including studies from Europe, United State and China.”

- Authors should also discuss the rise in the prevalence of Ab infections in the ICU during the COVID19 pandemic, which has been described in the literature

R: As you suggested, we discussed the rise in prevalence of AB infections during the Covid-19 pandemic page 18 , lines 391 -394 : “During the COVID-19 pandemic, there was a spike in AB health care-associated infections, primarily lower respiratory tract infections, in a number of ICU and non-ICU settings [35]. This AB outbreak inside the COVID-19 outbreak underlines the importance of appropriate prevention and control measures.”

A reference was also added.

Reference [35] Rangel K, Chagas TPG, De-Simone SG. Acinetobacter baumannii Infections in Times of COVID-19 Pandemic. Pathogens. 2021 Aug 10;10(8):1006. https://doi.org/10.3390/pathogens10081006. 

- Table1. Please add the unit of measure (ie years for age). Please add the first row with the total study population (n=155). I would not include azithromycin as an antiviral; rather, this is an antibiotic which has been used during COVID19 for its supposed action against SARS CoV2. The same in the text.

R: We added the units of measure and also the first row with the total study population. We totally agree with the reviewer about azithromycin. We corrected it in the table and in the text.

.- Table3. Please check numbers (VAP due to GN seems to be 15)

R: We checked the number of GN (n = 16). The mistake was the number of VAP due to pseudomonas (n = 3).

- Table4. Please check the row co-infection

R: We checked it. It is correct.

Reviewer #2: Dear Authors,

I commend your dedication to science and medicine in such a period of great strain for the critical care community. I read your paper with interest. I found it informative and valuable. Nevertheless, I have some comments for you. I think a minor revision is necessary to accept the paper on PLOS One.

R: We thank the reviewer for his/her efforts to review our manuscript and for his/her very positive comments.

Abstract:

Please add something regarding the statistical methods.

R: As you suggested a sentence was added about the statistical methods : Page 2 , Lines 33– 34 : “A multivariate regression analysis was performed to identify factors independently associated with mortality.”

Please reformulate the phrase "Death was associated with superinfection." The sample size and methods do not allow you to demonstrate any association. Instead, you may just say, "patients with superinfection showed a higher risk of death."

R : The phrase was reformulated as you required. Page 3, lines 46 -47.

Introduction:

Please introduce in the reference "Crit Care. 2022 Jun 13;26(1):176." regarding the increased risk of infection associated with corticosteroids.

R : The reference above was added in the section introduction as suggested.

Reference [14] Scaravilli V, Guzzardella A, Madotto F, Beltrama V, Muscatello A, Bellani G, et al. Impact of dexamethasone on the incidence of ventilator-associated pneumonia in mechanically ventilated COVID-19 patients: a propensity-matched cohort study. Crit Care. 2022 Jun 13;26(1):176. https://doi.org/10.1186/s13054-022-04049-2.

Methods: well done

Results: well done

Discussion: well done.

R: Thank you again for your comments.

Journal Requirements:

R: We have corrected the reference style.

R : We have Data Availability statement and added our data file: DATA CO.SUPERINF COVID ICU (Microsoft Excel)

---

## [Decision Letter · Decision Letter 1]

11 Nov 2022

What happened during COVID-19 in African ICUs? An observational study of pulmonary co-infections, superinfections, and mortality in Morocco.

PONE-D-22-23389R1

Dear Dr. Aissaoui,

We’re pleased to inform you that your manuscript has been judged scientifically suitable for publication and will be formally accepted for publication once it meets all outstanding technical requirements.

Kind regards,

SHUI YEE LEUNG

Academic Editor

PLOS ONE

Reviewers' comments:

Reviewer's Responses to Questions

**Comments to the Author**

1. If the authors have adequately addressed your comments raised in a previous round of review and you feel that this manuscript is now acceptable for publication, you may indicate that here to bypass the “Comments to the Author” section, enter your conflict of interest statement in the “Confidential to Editor” section, and submit your "Accept" recommendation.

Reviewer #1: All comments have been addressed

Reviewer #2: All comments have been addressed

2. Is the manuscript technically sound, and do the data support the conclusions?

Reviewer #1: Yes

Reviewer #2: Yes

3. Has the statistical analysis been performed appropriately and rigorously? 

Reviewer #1: Yes

Reviewer #2: Yes

4. Have the authors made all data underlying the findings in their manuscript fully available?

Reviewer #1: Yes

Reviewer #2: Yes

5. Is the manuscript presented in an intelligible fashion and written in standard English?

Reviewer #1: Yes

Reviewer #2: Yes

6. Review Comments to the Author

Reviewer #1: (No Response)

Reviewer #2: Dear Authors,

I do not have further comments. Best

Vittorio Scaravilli, MD

Fondazione IRCCS Ca' Granda - Ospedale Maggiore Policlinico, Milan, It

University of Milan, Milan, It.

7. PLOS authors have the option to publish the peer review history of their article (what does this mean?). If published, this will include your full peer review and any attached files.

Reviewer #1: No

Reviewer #2: **Yes: **Vittorio Scaravilli

---

## [Editor Report · Acceptance letter]

22 Nov 2022

PONE-D-22-23389R1 

What happened during COVID-19 in African ICUs? An observational study of pulmonary co-infections, superinfections, and mortality in Morocco. 

Dear Dr. Aissaoui:

I'm pleased to inform you that your manuscript has been deemed suitable for publication in PLOS ONE. Congratulations! Your manuscript is now with our production department. 

Kind regards, 

on behalf of

Dr. SHUI YEE LEUNG 

Academic Editor

PLOS ONE